# Paracoccidioidomycosis in people living with HIV/AIDS: A historical retrospective cohort study in a national reference center for infectious diseases, Rio de Janeiro, Brazil

**Eduardo Mastrangelo Falcão**[1][*], **Priscila Marques de Macedo**[1], **Dayvison Francis Saraiva Freitas**[1‡], **Andréa d'Avila Freitas**[2], **Beatriz Grinsztejn**[3‡], **Valdiléa Gonçalves Veloso**[3‡], **Rodrigo Almeida-Paes**[4‡], **Antonio Carlos Francesconi do Valle**[1]

**1** Clinical Research Laboratory on Infectious Dermatology, Evandro Chagas National Institute of Infectious Diseases, Fiocruz, Rio de Janeiro, Brazil, **2** Department of Inpatient Health Care, Evandro Chagas National Institute of Infectious Diseases, Fiocruz, Rio de Janeiro, Brazil, **3** Clinical Research Laboratory on HIV/AIDS, Evandro Chagas National Institute of Infectious Diseases, Fiocruz, Rio de Janeiro, Brazil, **4** Mycology Laboratory, Evandro Chagas National Institute of Infectious Diseases, Fiocruz, Rio de Janeiro, Brazil

☯ These authors contributed equally to this work.
‡ DFSF, BG, VGV and RA-P also contributed equally to this work.
* eduardo.falcao@ini.fiocruz.br

## Abstract

Paracoccidioidomycosis (PCM) is one of the main endemic systemic mycoses in Latin America, usually occurring in rural areas. When PCM occurs simultaneously with underlying immunosuppressive conditions, it can present as an opportunistic disease. Between 2000 and 2017, literature reported around 200 PCM cases in people living with HIV/AIDS (PLWHA). To address research gaps on this co-infection and to study its possible temporal changes in the last decade, we performed an active co-infection case search on the HIV/AIDS and PCM cohorts from a Brazilian reference center database from 1989 to 2019. We found 20 PLWHA among 684 PCM patients (2.92%), predominantly male (70.0%) and urban workers (80.0%). The median age of patients was higher in the 2010–2019 decade (p = 0.006). The occurrence of PCM in PLWHA was lower when compared with other fungal diseases. Although 50.0% of the patients had already been diagnosed with HIV infection and presented CD4+ T cell counts greater than 200/mm$^3$ at the time of PCM diagnosis, the suspicion of immunosuppression in the context of atypical and more severe clinical forms of PCM revealed the diagnosis of HIV infection in 35.0% of the patients. Two (10.0%) patients had an evolution compatible with immune reconstitution inflammatory syndrome (IRIS) after starting antiretroviral therapy (ART).We highlight the importance of considering a PCM diagnosis in PLWHA to prevent a late-onset treatment and progression to severe manifestations and unfavorable outcomes. In addition, HIV investigation is recommended in PCM patients, especially those with atypical and more severe clinical presentations.

**Data Availability Statement:** All relevant data are within the manuscript and its Supporting Information files.

**Funding:** The work received financial support from the Evandro Chagas National Institute of Infectious Diseases, Oswaldo Cruz Foundation (INI/ Fiocruz), which provided infrastructure and paid for publishing expenses. The funders had no role in study design, data collection and analysis, decision to publish, or preparation of the manuscript.

**Competing interests:** The authors have declared that no competing interests exist

## Author summary

Paracoccidioidomycosis (PCM) is a severe systemic mycosis caused by inhalation of fungi belonging to the genus *Paracoccidioides* present in the soil of endemic areas in Latin America. However, it is still a neglected disease, affecting vulnerable populations such as rural workers. In the last decade, there was an increase of acute PCM cases in young people living in urban areas of the endemic area of Rio de Janeiro, Brazil. This could increase the occurrence of PCM in people living HIV/AIDS (PLWHA) because they are more concentrated in these regions. When PCM and immunosuppression due to AIDS occur simultaneously, PCM can present as an opportunistic disease, with more severe, invasive, and atypical presentations. In these cases, late diagnosis and treatment can lead to higher risk of complications, sequelae, and deaths. PCM occurrence in PLWHA is scarcely reported in the literature. This study aims to describe the clinical profile of patients diagnosed with PCM and HIV co-infection from a 30-year historical cohort followed at a Brazilian reference center for infectious diseases. Our results revealed that the suspicion of this co-infection in patients with more severe clinical forms of PCM as well as routine HIV testing in PCM patients could help to prevent late-onset treatment and progression to unfavorable outcomes.

## Introduction

Paracoccidioidomycosis (PCM) is one of the main endemic systemic mycoses in Latin America; about 80% of the region's PCM cases occur in Brazil. Infection occurs through inhalation of *Paracoccidioides* spp. Usually, after activities involving soil management in rural areas. It is estimated that around 10 million people are infected in South America, and up to 2% will develop PCM symptoms. Most will progress to disease years after infection, presenting mainly lung disease (chronic form). PCM can also occur in the acute form, less frequently, especially in young people, affecting the mononuclear phagocytic system and spreading rapidly to multiple organs [1, 2].

The ability to control *Paracoccidioides* spp. is related to an effective cellular immune response resulting in the formation of compact granulomas [3]. When PCM and immunosuppression occur simultaneously, a change in the natural history of this mycosis may occur, and PCM presents as an opportunistic disease. In these cases, the fungal disease may develop acutely, even years after infection, or with mixed clinical aspects of acute and chronic forms, hampering the clinical classification [4–7].

In PCM endemic areas, the estimated prevalence of individuals infected with *Paracoccidioides* spp. living with HIV is higher than 12%. Although their risk for developing PCM symptoms is not well established, primary prophylaxis for PCM in PLWHA is not routinely recommended [8]. Data from the World Health Organization (WHO) showed 37.7 million people living with HIV infection worldwide in 2020 [9]. In Brazil, from 1980 to 2021, 1,045,355 cases of AIDS were identified [10]. Until 1995, only 27 cases of PCM had been described among at least 500,000 people living with HIV infection in South America [11]. Until 2019, the total number of reported PCM cases in PLWHA was lower than 200 [11–13].

This study aimed to describe the clinical, epidemiological, and laboratory aspects of a cohort of patients with PCM and living with HIV in a reference center for infectious diseases in the Rio de Janeiro state, an important PCM endemic area in Brazil, contributing to the knowledge on this uncommon but serious co-infection. Moreover, a comparison between patients diagnosed between 1989–2009 and 2010–2019 was conducted, to evaluate possible temporal changes in epidemiologic and clinic characteristics of this co-infection over the 2 last decades.

## Methods

### Ethics statement

The Research Ethics Committee of the Evandro Chagas National Institute of Infectious Diseases (INI/FIOCRUZ), a reference center for PCM and HIV/AIDS in Rio de Janeiro State, Brazil approved this study (appreciation numbers 42590515.0.0000.5262 and 26066619.0.0000.5262).

### Study design

This is a historical retrospective cohort study developed at the Evandro Chagas National Institute of Infectious Diseases (INI/FIOCRUZ), a reference center for PCM and HIV/AIDS in Rio de Janeiro State, Brazil.

### Patients

We performed an active case search, on the HIV/AIDS and PCM cohorts from the INI/FIOCRUZ database, from 1989 to 2019. After inclusion, the medical records were deidentified to protect patients' privacy. Inclusion criteria were the diagnosis of PCM according to the Brazilian Consensus on PCM [1] and HIV infection as determined by the Brazilian Ministry of Health [10]. Active and regular contact with the absent patients, monitoring the delivery of medication free of charge, as well as optimization for treatment regimens and medical appointment scheduling were strategies used to prevent loss to follow-up.

### Data analysis

The variables analyzed were socio-demographic: gender, age, city of residence, occupation; clinical: time between HIV and PCM diagnosis, clinical form of PCM according to the Brazilian Consensus on PCM [1], affected organs, co-infections; laboratorial: TCD4+ count and viral load at the moment of PCM diagnosis (or at the moment of HIV diagnosis when it occurred after PCM), diagnostic method for PCM, titer of specific antibodies against *Paracoccidioides* spp. (double radial immunodiffusion–DID) [14] before initiation of antifungal therapy; and therapeutic: drugs used to treat PCM, time of PCM treatment, and adherence to HIV and PCM treatments.

### Statistical analysis

The patients were divided into two groups, based on the time of PCM diagnosis (before and after 2010), to enable the analysis of temporal evolution, comparing clinical and epidemiological changes of the two diseases over the time, and to verify whether advances in the diagnosis and therapy of HIV infection had an impact on the presentation of PCM in PLWHA. Statistical analyses were performed with the R program (version 4.0.5), using the Mann-Whitney test for quantitative variables and the Fisher test for qualitative, considering significance levels lower than 0.05.

## Results

### Epidemiological data

The search for patients in the PCM cohort of INI/FIOCRUZ revealed 684 patients assisted from 1989 to 2019. Twenty patients from this cohort living with HIV were included in this study and divided into two groups. The first group (A) comprises eight patients with PCM

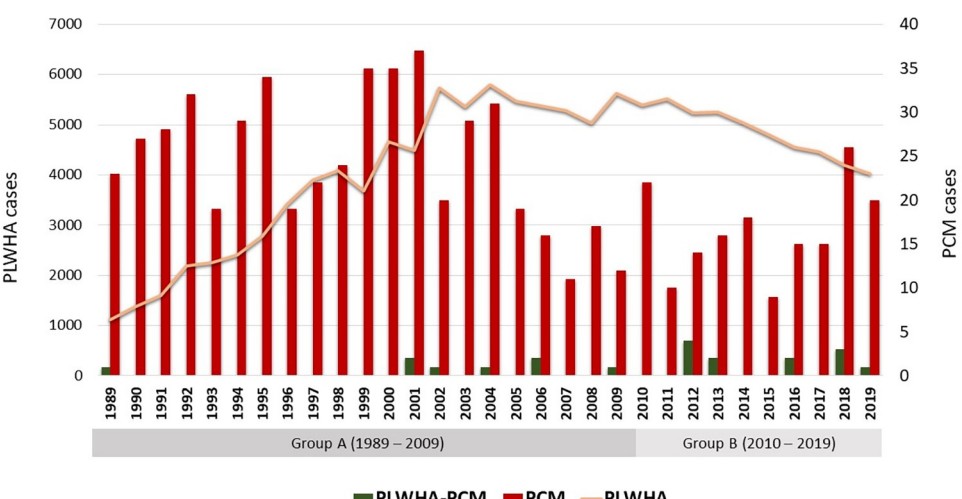

**Fig 1. Annual cases of people living with HIV/AIDS at Rio de Janeiro State, PCM at INI/FIOCRUZ, and association of PCM and HIV/AIDS at INI/FIOCRUZ between 1989 and 2019.** PLWHA: People living with HIV/AIDS; PCM: Paracoccidioidomycosis. Source: DATASUS and SIPEC-FIOCRUZ.

diagnosed between 1989 and 2009, and the second group (B) twelve patients diagnosed with PCM between 2010 and 2019. Fig 1 shows the distribution of the cases over the years.

Gender, residence and occupation did not present a statistically significant difference between groups A and B. The median age in group A was 36 years (range: 25–51), while in group B it was 50 years (range: 33–63). This difference was statistically significant (p = 0.006). Table 1 describes the main epidemiological characteristics of the patients from this study.

## Clinical and laboratory data

Table 2 presents clinical data of HIV and PCM infection. The suspicion of immunosuppression in patients with atypical and more severe clinical forms of PCM (Fig 2) revealed the diagnosis of HIV infection in 37.5% of patients in group A and 33.3% in group B. In group A, one patient was diagnosed with HIV infection after 11 years when PCM relapsed requiring new

**Table 1. Epidemiological characteristics of the cases analyzed in this study.**

| Variables | Group A (1989–2009) | | Group B (2010–2019) | |
|---|---|---|---|---|
| | N | % (CI) | N | % (CI) |
| **Gender** | | | | |
| Male | 5 | 62.5 (24.5–91.5) | 9 | 75.0 (42.8–94.5) |
| Female | 3 | 37.5 (8.5–75.5) | 3 | 25.0 (5.5–57.2) |
| **Residence** | | | | |
| Rio de Janeiro (capital) | 3 | 37.5 (8.5–75.5) | 5 | 41.7 (15.2–72.3) |
| Metropolitan municipalities | 4 | 50.0 (15.7–84.3) | 5 | 41.7 (15.2–72.3) |
| Other municipalities (countryside) | 1 | 12.5 (0.3–52.7) | 2 | 16.7 (2.1–48.4) |
| **Occupation** | | | | |
| Rural worker | 2 | 25.0 (3.2–65.1) | 1 | 8.3 (0.2–38.5) |
| Urban worker | 5 | 62.5 (24.5–91.5) | 11 | 91.7 (61.5–99.8) |
| NA | 1 | 12.5 (0.3–52.7) | 0 | 0.0 (0.0–26.5) |

NA: Not available; CI: 95% confidence interval.

**Table 2. Data of the HIV and PCM infection from the cases analyzed in this study.**

| Variables | Group A (1989–2009) | | Group B (2010–2019) | |
|---|---|---|---|---|
| | N | % (CI) | N | % (CI) |
| **HIV diagnosis** | | | | |
| Before PCM diagnosis | 2 | 25.0 (3.2–65.1) | 8 | 66.7 (34.9–90.1) |
| Coincidental | 3 | 37.5 (8.5–75.5) | 4 | 33.3 (9.9–65.1) |
| After PCM diagnosis | 3 | 37.5 (8.5–75.5) | 0 | 0.0 (0.0–26.5) |
| **CD4 cells count (cells/mm$^3$)** | | | | |
| >200 | 4 | 50.0 (15.7–84.3) | 7 | 58.3 (27.7–84.8) |
| <200 | 2 | 25.0 (3.2–65.1) | 4 | 33.3 (9.9–65.1) |
| NA | 2 | 25.0 (3.2–65.1) | 1 | 8.3 (0.2–38.5) |
| **HIV viral load (copies/mL)** | | | | |
| >100,000 | 0 | 0.0 (0.0–37.0) | 2 | 16.7 (2.1–48.4) |
| 10,000–100,000 | 3 | 37.5 (8.5–75.5) | 2 | 16.7 (2.1–48.4) |
| 50–9,999 | 3 | 37.5 (8.5–75.5) | 3 | 25.0 (5.5–57.2) |
| <50 | 0 | 0.0 (0.0–37.0) | 3 | 25.0 (5.5–57.2) |
| NA | 2 | 25.0 (3.2–65.1) | 2 | 16.7 (2.1–48.4) |
| **PCM clinical form** | | | | |
| Acute/subacute | 2 | 25.0 (3.2–65.1) | 4 | 33.3 (9.9–65.1) |
| Chronic | 5 | 62.5 (24.5–91.5) | 6 | 50.0 (21.1–78.9) |
| Mixed | 1 | 12.5 (0.3–52.7) | 2 | 16.7 (2.1–48.4) |
| **PCM affected organs[a]** | | | | |
| Aerodigestive tract mucosa | 5 | 62.5 (24.5–91.5) | 9 | 75.0 (42.8–94.5) |
| Lymph nodes | 5 | 62.5 (24.5–91.5) | 6 | 50.0 (21.1–78.9) |
| Skin | 3 | 37.5 (8.5–75.5) | 6 | 50.0 (21.1–78.9) |
| Lungs | 4 | 50.0 (15.7–84.3) | 5 | 41.7 (15.2–72.3) |
| Adrenal | 0 | 0.0 (0.0–37.0) | 1 | 8.3 (0.2–38.5) |
| Bones | 0 | 0.0 (0.0–37.0) | 1 | 8.3 (0.2–38.5) |
| Spleen | 0 | 0.0 (0.0–37.0) | 1 | 8.3 (0.2–38.5) |
| **PCM serology (DID)** | | | | |
| Reactive | 5 | 62.5 (24.5–91.5) | 11 | 91.7 (61.5–99.8) |
| Non-reactive | 2 | 25.0 (3.2–65.1) | 1 | 8.3 (0.2–38.5) |
| NA | 1 | 12.5 (0.3–52.7) | 0 | 0.0 (0.0–26.5) |
| **PCM treatment[b]** | | | | |
| Sulfamethoxazole/trimethoprim | 8 | 100.0 (63.0–100.0) | 9 | 75.0 (42.8–94.5) |
| Amphotericin B | 3 | 37.5 (8.5–75.5) | 3 | 25.0 (5.5–57.2) |
| Itraconazole | 0 | 0.0 (0.0–37.0) | 8 | 66.7 (34.9–90.1) |
| Fluconazole | 1 | 12.5 (0.3–52.7) | 8 | 66.7 (34.9–90.1) |
| Ketoconazole | 1 | 12.5 (0.3–52.7) | 0 | 0.0 (0.0–26.5) |
| **Treatment adherence** | | | | |
| Good | 3 | 37.5 (8.5–75.5) | 8 | 66.7 (34.9–90.1) |
| Poor | 5 | 62.5 (24.5–91.5) | 4 | 33.3 (9.9–65.1) |
| **Other co-infections[c]** | | | | |
| Tuberculosis | 4 | 50.0 (15.7–84.3) | 3 | 25.0 (5.5–57.2) |
| Syphilis | 1 | 12.5 (0.3–52.7) | 1 | 8.3 (0.2–38.5) |
| Esophageal candidiasis | 0 | 0.0 (0.0–37.0) | 2 | 16.7 (2.1–48.4) |
| Cryptococcosis | 2 | 25.0 (3.2–65.1) | 0 | 0.0 (0.0–26.5) |
| Pneumocystosis | 1 | 12.5 (0.3–52.7) | 0 | 0.0 (0.0–26.5) |

(*Continued*)

**Table 2.** (Continued)

| Variables | Group A (1989–2009) | | Group B (2010–2019) | |
|---|---|---|---|---|
| | N | % (CI) | N | % (CI) |
| Kaposi´s sarcoma | 0 | 0.0 (0.0–37.0) | 1 | 8.3 (0.2–38.5) |

NA: not available; PCM: paracoccidioidomycosis; DID: double radial immunodiffusion; CI: 95% confidence interval.

[a]Eighteen patients (90.0%) had multifocal PCM.

[b]Fourteen patients (70.0%) needed more than one drug.

treatment. In group B, 66.7% of patients had already been diagnosed with HIV infection before PCM diagnosis, most (62.5%) undergoing regular treatment. Fifty percent of the patients in both groups had CD4+ T cell counts greater than 200/mm$^3$ at the time of diagnosis of PCM (median: group A = 491 cells/mm3 and group B = 236 cells/mm3, p-value = 0.18).

Two patients of group B progressed with clinical signs compatible with IRIS: a 33 year-old man and a 54 year-old woman. Both had TCD4+ cell count below 200 cells/mm$^3$ at the time of diagnosis (188 and 93 cells/mm$^3$, respectively) and presented enlarged fistulized lymph nodes and fever associated with the increased TCD4+ cell count, 2 and 4 months after starting ART, respectively. Systemic corticosteroid was necessary for three months for the first patient with clinical improvement. Both patients evolved with cure of PCM. (Fig 3).

The PCM most affected organs in both groups were the mucosa of the upper airways (group A = 62.5% and group B = 75.0%), the lymph nodes (62.5% and 50.0%, respectively), and the skin (37.5% and 50.0%, respectively). The clinical specimens used for the PCM

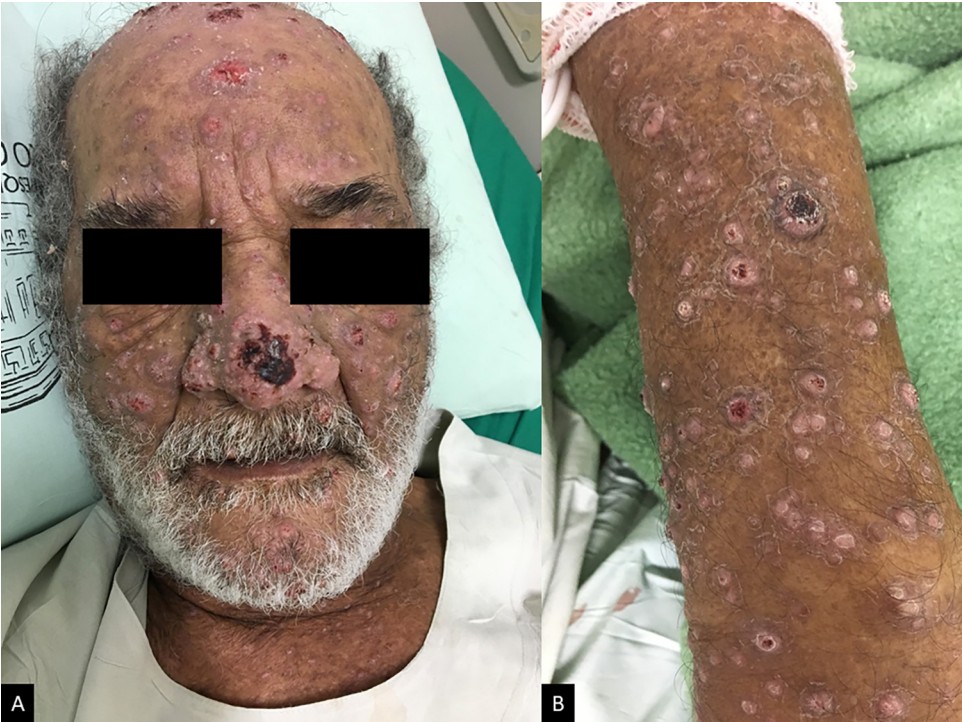

**Fig 2. Cutaneous lesions due to hematogenous dissemination of *Paracoccidioides* sp. in a patient with HIV/aids.**
Patient from group B, 63 years old, presenting (A) ulcerative and ulcerous-vegetative lesions on the face, (B) papules and ulcerative lesions covered by crusts on the forearm.

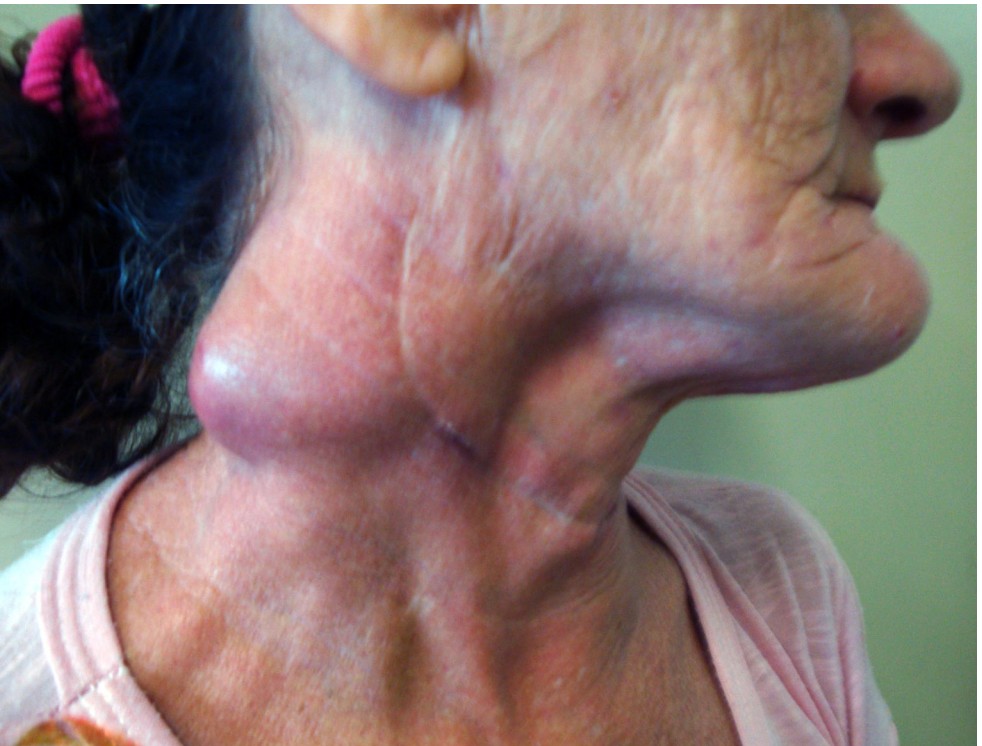

**Fig 3. Bulky lymphadenopathy due to hematogenous dissemination of *Paracoccidioides* sp. in a patient living with HIV/AIDS.** Patient from group B, 54 years old, presenting a massive enlargement of cervical lymph nodes in the context of immune reconstitution inflammatory syndrome (IRIS).

diagnosis were obtained from skin or mucosal lesions in 70.0% of the patients and from lymph nodes in 15.0%. Histopathological examination was the main method for diagnosis, performed in 65.0% of the patients, and all revealed fungal elements suggestive of *Paracoccidioides* spp. Mycological examination (direct examination with 10% potassium hydroxide and/or fungal culture) was performed in 60.0% of patients and was positive in 66.6% of them. One patient (5.0%) was diagnosed serologically (DID titer 1:64).

PCM treatment ranged from 24 to 60 months, (median 30 months) and from 12 to 68 months (median 29 months (p-value = 0.72) in groups A and B, respectively. Sulfamethoxazole-trimethoprim (SMZ-TMP) was used for the treatment of PCM in all patients from group A and in 75.0% of patients from group B. Another drug was added or replaced the initial treatment due to poor response (50.0% in group A and 66.7% in group B) or drug interaction (16.7% in group B). Antifungals used were ketoconazole, fluconazole or amphotericin B in group A and amphotericin B, itraconazole and fluconazole in group B.

Poor adherence to treatment for both infections led to more severe conditions and recurrences (Fig 4). In group A, 62.5% used the medications irregularly, while in group B, adherence was poor in 33.3%. Most patients in group A (62.5%) were diagnosed with another endemic or opportunistic infection during treatment, including tuberculosis (50.0%), syphilis (12.5%), and cryptococcosis (25.0%). In group B, other infections were present in 33.3% of patients, including tuberculosis (25.0%), esophageal candidiasis (16.7%), and syphilis (8.3%).

There were a total of five deaths in group A, four patients (50.0%) died due to neurocryptococcosis (n = 1), pseudotumoral brain lesion (n = 1), sepsis (n = 1), and meningitis without defined etiology (n = 1). In group B, one patient (8.3%) died in another hospital from an

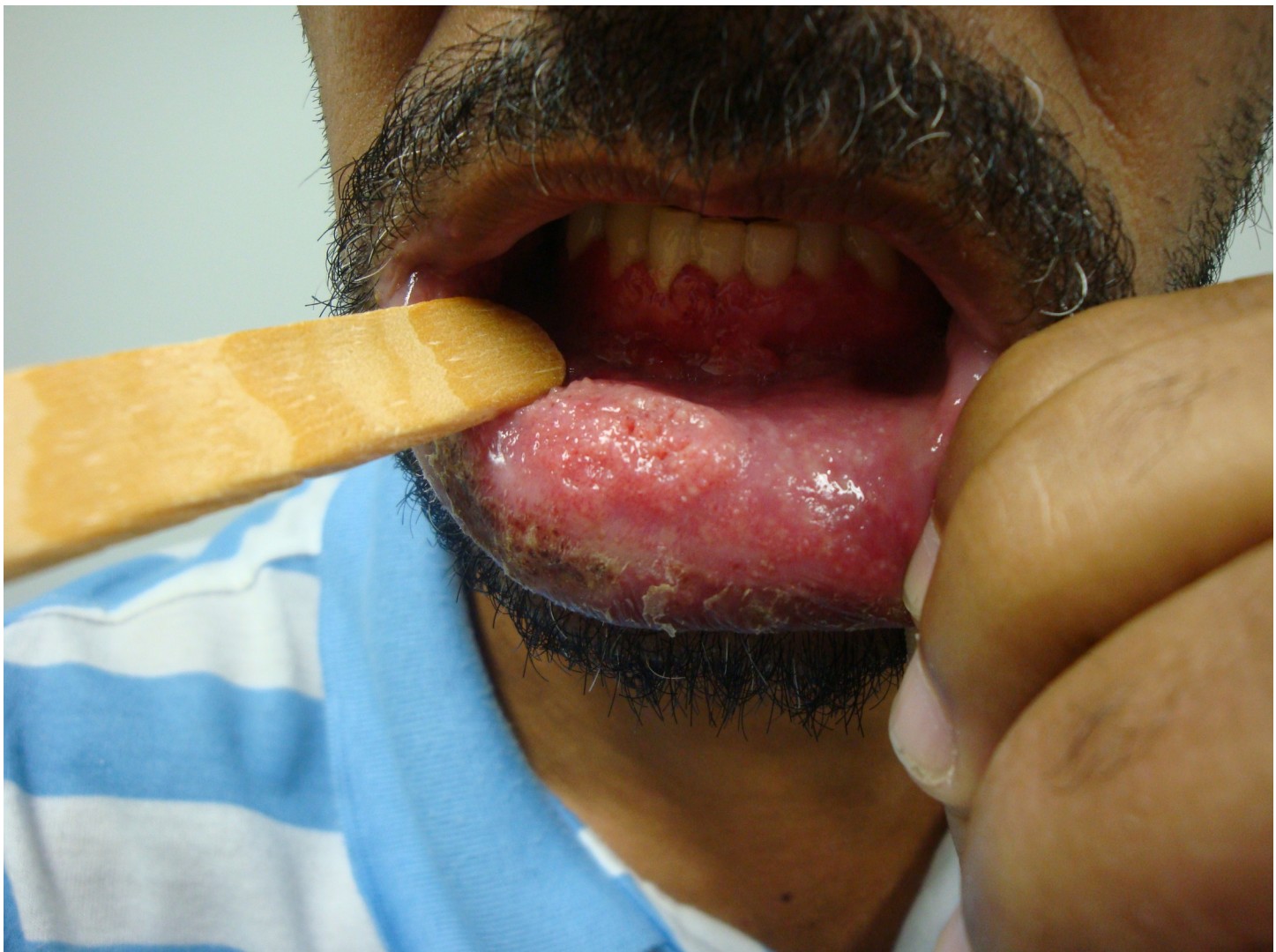

**Fig 4. Ulcero-vegetative lesion on the lower lip due to hematogenous dissemination of *Paracoccidioides* sp. in a patient living with HIV/AIDS.** Patient from group B, 45 years old, with recurrent oral lesion after two years of an irregular treatment.

unknown cause. Two months earlier, on his last visit at INI/FIOCRUZ, he presented a TCD4 + cell count of 129 cells/mm3 and no PCM complications.

## Discussion

Fungi are major contributors to opportunistic infections and deaths related to HIV/AIDS. The main fungal diseases in patients living with HIV are pneumocystosis, cryptococcosis, and histoplasmosis [15, 16]. In our center, sporotrichosis has also assumed an important role as an opportunistic disease among these patients, reaching levels close to other opportunistic fungal diseases [17]. This study has shown that, despite increasing PCM cases in urban areas of Rio de Janeiro, as it happened with sporotrichosis in the early 2000s, PCM/HIV co-infection seems to be not currently following the same dynamics as sporotrichosis/HIV co-infection.

At INI/FIOCRUZ, there is a historical cohort of patients with PCM since 1960 with approximately 1,200 cases. In this cohort, the occurrence of PCM in PLWH was lower when

compared with other fungal diseases, which is similar to the literature data [18]. Two factors may contribute to these observations. First, in Brazil, individuals with low TCD4+ cell levels ($< 100$ cells/mm$^3$) receive sulfonamides for pneumocystosis prophylaxis [10], an effective medication against *Paracoccidioides* spp. that may provide protection against the occurrence of clinical manifestations of PCM. Second, while PCM has a rural epidemiological profile, HIV predominates in urban areas [10].

PCM-HIV co-infection is reported predominantly in urban areas, affecting younger individuals and in a smaller male to female ratio [12, 13, 19], as observed in our study. However, in group A at least 25.0% of the patients had worked in rural areas, while in group B, this frequency was only 8.3%. Although not significant, perhaps due to the low number of patients in both groups, it is possible that this change is associated with recent modifications in the epidemiological profile of PCM in the state of Rio de Janeiro, characterized by an increase of acute forms in urban areas associated with the construction of a highroad in the Rio de Janeiro metropolitan area [20]. The number of co-infection cases was not affected by the PCM urbanization. However, the implementation of a compulsory notification and epidemiological surveillance, which is already in progress [21], is of great importance considering the possibility of a greater number of cases in the future.

The improved survival of people living with HIV infection observed in the post-highly active ART era can help explain the higher age in group B patients. The first drugs for HIV treatment started to be provided by the Brazilian Ministry of Health in 1991, but a wide free supply in Brazil was only available in 1996 [22]. In 2013 the test-and-treat strategy was adopted by the Ministry of Health [10]. The predominance of male individuals in both groups is similar to the epidemiological characteristics of HIV/AIDS in Brazil and the profile of PCM in which men are more affected [1, 10].

In this study, the chronic form, although predominant, was observed in a lower frequency than expected. This can be explained by the opportunistic behavior of PCM in individuals with AIDS, usually more severe and invasive, combining clinical signs of the acute and the chronic forms (mixed form), or presenting major symptoms of the acute form as observed in 25 to 33.3% of the cases herein analyzed. These results are equivalent to those described in some hyperendemic regions, where acute PCM can correspond to up to 26% of the PCM cases [23].

Late diagnosis and treatment for patients living with HIV, especially those with TCD4+ cell counts below 50/mm$^3$, are important risk factors for IRIS. This causes a paradoxical worsening of preexisting untreated opportunistic infections [24]. The worsening of PCM symptoms after the onset of ART has been reported in the literature [25]. Therefore, IRIS should be considered when managing these patients after excluding other factors that may be related to poor prognosis [26, 27]. Although uncommon, a paradoxical reaction not associated with immunosuppressive conditions may also occur in PCM treatment. The two patients herein reported with IRIS did not present any factor that could explain the abrupt clinical worsening other than the start of ART. The use of corticosteroids seems to be helpful in these cases [24, 28], as observed in one of our IRIS cases with clinical deterioration and poor response to antifungal treatment.

As the fungal isolates of most patients herein studied were not available, we could not identify the *Paracoccidioides* species. However, *P. brasiliensis sensu stricto* is reported to be the most prevalent species in the state of Rio de Janeiro, both in non-HIV and HIV groups [29, 30]. The high virulence of this species [31] may also have contributed to the greater severity observed in some patients of this study.

Concerning the immunological aspects in PCM diagnosis, AIDS-related immunosuppression constitutes a critical issue to be discussed. The reduced antibody production to lower than the DID detection threshold may lead to false-negative results more frequently in

individuals with AIDS [11, 13, 23]. The DID, immunoblotting (IB), counter immunoelectrophoresis (CIE), and enzyme-linked immunosorbent assay (ELISA) are used in reference centers. Their sensitivity and specificity vary according to the test, the antigen used, and the fungal species responsible for the infection [1]. False-negative results in DID can occur in up to 28.5% and 7.4% of the acute and chronic PCM cases, respectively, usually corresponding to unifocal or mild presentations [32].

On the other hand, the positivity of DID, especially in higher titers, is strongly associated with PCM activity and can be very helpful in the diagnosis [1]. For example, one patient from group A had respiratory manifestations associated with a high PCM DID titer and the absence of laboratorial tuberculosis which led to the initiation of treatment for PCM followed by a good clinical response. Besides the increased risk in patients living with HIV/AIDS, some other endemic and opportunistic infections may also have a higher incidence in patients with PCM. It has been reported that the rate of PCM patients with tuberculosis co-infection may be as high as 20% [1]. The higher incidence of PCM and tuberculosis co-infection in people living with HIV in our cohort (50.0% in group A and 25.0% in group B) reflects the clinical profile of immunosuppression with multiple opportunistic diseases. We herein detail a group B patient who presented *Paracoccidioides* spp. structures on the histopathological examination of cervical lymph nodes and a detectable level of *Mycobacterium tuberculosis* DNA in the same sample. Both conditions were treated simultaneously, with a good outcome.

Regarding the therapeutic aspects of PCM in individuals with HIV, the broad spectrum of pharmacological interactions of itraconazole, the first choice for the treatment of PCM, makes SMZ-TMP a frequently used alternative therapy. This was the most frequent therapeutic choice in our series. As a disadvantage, its dosage requiring drug administration twice a day makes adherence difficult. This was observed in both groups, especially group A, in which poor adherence occurred almost twice as often as in group B. Poor adherence to long-term treatments of chronic diseases is typically observed in PLWHA, and the low adherence to ART can lead to drug resistance and consequently to a worse prognosis [33]. Low adherence to PCM treatment is usually associated with complications, relapses, sequelae, and deaths [1]. In this study, the clinical severity as well as the low compliance with treatment observed in some cases contributed to the difficulty in the therapeutic management in these cases, leading to a longer duration and wide variation in the treatment time. One patient from group B presented a recurrence of an oral lesion after two years of irregular treatment with SMZ-TMP (Fig 4). The lesion persisted 20 months after reintroduction of SMZ-TMP, and hospitalization was necessary for administering amphotericin B. After discharge, itraconazole was prescribed for 48 months, but the patient abandoned follow-up. Although the oral lesion had already healed, he presented a new skin lesion in the mandibular region on his last visit. The irregular use of SMZ-TMP for PCM treatment combined with a previous long period of prophylaxis for *Pneumocystis jirovecii* may also have led to drug resistance in this case. Another patient from group B presented perforation of the palate as a sequel, probably due to the late onset of treatment.

Deaths in both groups were not directly related to PCM. The reduction in mortality in group B compared to group A, although not statistically significant, may be due to a more effective diagnosis and treatment of HIV/AIDS over the time. Deaths by PCM in PLWHA are reported to occur early, before antifungal treatment, and mortality seems to be similar to patients without co-infections [19].

In this context, early treatment and adherence-enhancing interventions such as active search, availability of therapeutic options with a comfortable regimen, lower rate of adverse events, as well as a good patient-provider relationship are fundamental to improve the quality of life and favorable outcomes of both infections.

## Conclusions

A diagnosis of PCM in PLWHA should be considered to prevent late-onset treatment and the progression to severe manifestations and unfavorable outcomes. In addition, HIV investigation is recommended in PCM patients, especially those with atypical and more severe clinical presentations. Mycological examination and immunodiagnostic techniques with a better sensitivity and specificity profile such as IB should be available in endemic areas to promote the early diagnosis of these patients.

Although in this study we did not detect a significant increase in the number of cases in the second period studied, which includes the moment when epidemiological alterations occurred in the state of Rio de Janeiro, it is essential to implement surveillance strategies to observe whether this change may impact the occurrence of PCM in vulnerable groups in the urban environment over time.

## Supporting information

**S1 Strobe Checklist. STROBE Statement.**
(DOCX)

## Acknowledgments

The authors are sincerely grateful to Dr. Maria Clara Gutierrez-Galhardo for critical reading of the manuscript.

## Author Contributions

**Conceptualization:** Eduardo Mastrangelo Falcão, Priscila Marques de Macedo, Antonio Carlos Francesconi do Valle.

**Data curation:** Eduardo Mastrangelo Falcão, Priscila Marques de Macedo, Andréa d'Avila Freitas, Antonio Carlos Francesconi do Valle.

**Formal analysis:** Eduardo Mastrangelo Falcão, Priscila Marques de Macedo, Dayvison Francis Saraiva Freitas, Andréa d'Avila Freitas, Beatriz Grinsztejn, Valdiléa Gonçalves Veloso, Rodrigo Almeida-Paes, Antonio Carlos Francesconi do Valle.

**Investigation:** Eduardo Mastrangelo Falcão, Priscila Marques de Macedo, Dayvison Francis Saraiva Freitas, Andréa d'Avila Freitas, Beatriz Grinsztejn, Valdiléa Gonçalves Veloso, Rodrigo Almeida-Paes, Antonio Carlos Francesconi do Valle.

**Methodology:** Eduardo Mastrangelo Falcão, Priscila Marques de Macedo, Dayvison Francis Saraiva Freitas, Andréa d'Avila Freitas, Beatriz Grinsztejn, Valdiléa Gonçalves Veloso, Rodrigo Almeida-Paes, Antonio Carlos Francesconi do Valle.

**Project administration:** Antonio Carlos Francesconi do Valle.

**Resources:** Beatriz Grinsztejn, Valdiléa Gonçalves Veloso, Rodrigo Almeida-Paes, Antonio Carlos Francesconi do Valle.

**Software:** Eduardo Mastrangelo Falcão, Dayvison Francis Saraiva Freitas, Rodrigo Almeida-Paes, Antonio Carlos Francesconi do Valle.

**Supervision:** Priscila Marques de Macedo, Dayvison Francis Saraiva Freitas, Beatriz Grinsztejn, Valiléa Gonçalves Veloso, Antonio Carlos Francesconi do Valle.

**Validation:** Eduardo Mastrangelo Falcão, Priscila Marques de Macedo, Dayvison Francis Saraiva Freitas, Andréa d'Avila Freitas, Beatriz Grinsztejn, Valdiléa Gonçalves Veloso, Rodrigo Almeida-Paes, Antonio Carlos Francesconi do Valle.

**Visualization:** Eduardo Mastrangelo Falcão, Priscila Marques de Macedo, Dayvison Francis Saraiva Freitas, Andréa d'Avila Freitas, Beatriz Grinsztejn, Valdiléa Gonçalves Veloso, Rodrigo Almeida-Paes, Antonio Carlos Francesconi do Valle.

**Writing – original draft:** Eduardo Mastrangelo Falcão, Priscila Marques de Macedo, Dayvison Francis Saraiva Freitas, Andréa d'Avila Freitas, Beatriz Grinsztejn, Valdiléa Gonçalves Veloso, Rodrigo Almeida-Paes, Antonio Carlos Francesconi do Valle.

**Writing – review & editing:** Eduardo Mastrangelo Falcão, Priscila Marques de Macedo, Dayvison Francis Saraiva Freitas, Andréa d'Avila Freitas, Beatriz Grinsztejn, Valdiléa Gonçalves Veloso, Rodrigo Almeida-Paes, Antonio Carlos Francesconi do Valle.

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
