## [Decision Letter · Decision Letter 0]

8 Apr 2022

Dear Dr Falcao,

Thank you very much for submitting your manuscript "Paracoccidioidomycosis in people living with HIV/AIDS: a historical retrospective cohort study in a national reference center for infectious diseases, Rio de Janeiro, Brazil" for consideration at PLOS Neglected Tropical Diseases. As with all papers reviewed by the journal, your manuscript was reviewed by members of the editorial board and by several independent reviewers. In light of the reviews (below this email), we would like to invite the resubmission of a significantly-revised version that takes into account the reviewers' comments. 

We cannot make any decision about publication until we have seen the revised manuscript and your response to the reviewers' comments. Your revised manuscript is also likely to be sent to reviewers for further evaluation.

Sincerely,

Ahmed Fahal, FRCS, FRCSI, FRCSG, MS, MD, FRCP(London)

Deputy Editor

Ahmed Fahal

Deputy Editor

Reviewer's Responses to Questions

**Key Review Criteria Required for Acceptance?**

**Methods**

-Are the objectives of the study clearly articulated with a clear testable hypothesis stated?

-Is the study design appropriate to address the stated objectives?

-Is the population clearly described and appropriate for the hypothesis being tested?

-Is the sample size sufficient to ensure adequate power to address the hypothesis being tested?

-Were correct statistical analysis used to support conclusions?

-Are there concerns about ethical or regulatory requirements being met?

Reviewer #1: Are the objectives of the study clearly articulated with a clear testable hypothesis stated? No

Is the study design appropriate to address the stated objectives? Yes

Is the population clearly described and appropriate for the hypothesis being tested. Yes

Is the sample size sufficient to ensure adequate power to address the hypothesis being tested? No

Were correct statistical analysis used to support conclusions? No

Are there concerns about ethical or regulatory requirements being met? No

**Results**

-Does the analysis presented match the analysis plan?

-Are the results clearly and completely presented?

-Are the figures (Tables, Images) of sufficient quality for clarity?

Reviewer #1: Does the analysis presented match the analysis plan? Yes

Are the results clearly and completely presented? Yes

Are the figures (Tables, Images) of sufficient quality for clarity? Yes

**Conclusions**

-Are the conclusions supported by the data presented?

-Are the limitations of analysis clearly described?

-Do the authors discuss how these data can be helpful to advance our understanding of the topic under study?

-Is public health relevance addressed?

Reviewer #1: Are the conclusions supported by the data presented? No

-Are the limitations of analysis clearly described? No

Do the authors discuss how these data can be helpful to advance our understanding of the topic under study? No

Is public health relevance addressed?Yes

**Editorial and Data Presentation Modifications?**

Reviewer #1: (No Response)

**Summary and General Comments**

Reviewer #1: 1-The abstract needs to be improved and must include more data related with the study.

2-The Key words were not described.

3-The Results: They are very extensive, repetitive in both text and tables.

 4-The Tables could be condensed in one or two.

5- the presented figures, maybe the number two has more relevance to be included into the paper. the others could be removed.

 6-The discussion section is very extensive, few adequated and needs to be more concise and focalized on HIV/ PCM coinfection together with the literature data.

7-Generalizations are used with frequency and they must be avoided, due to the small size of the individuals included.

Minor comments.

1-Please verify reference number 12. It is inadequate for citation. See Almeida et al 2017.Benard et al 2000. Some of the references are few originals 

 2-The paragraph between lines 87-90 is inadeaquate.

 3-There is a contradiction in values related with the prevalence of the acute clinical form of PCM <10% line 281, and 26% line288.

 4-It would be interesting to cite the reference whic describe 12% of PCM in HIV infected patients Line 90.

5-It is very importan to clarify why the high variability of therapy time in these patients From 12 to 60 months?This is very different from elsewere 

 6-I suggest to improve the information and disccussion of the two cases who developed IRIS. This is an exciting matter to give an upgrade in your manuscript.

 7-The English style must be reviewed.

PLOS authors have the option to publish the peer review history of their article (what does this mean?). If published, this will include your full peer review and any attached files.

Reviewer #1: No
---

## [Decision Letter · Decision Letter 1]

20 May 2022

Dear Dr Falcao,

We are pleased to inform you that your manuscript 'Paracoccidioidomycosis in people living with HIV/AIDS: a historical retrospective cohort study in a national reference center for infectious diseases, Rio de Janeiro, Brazil' has been provisionally accepted for publication in PLOS Neglected Tropical Diseases.

Best regards,

Ahmed Fahal, FRCS, FRCSI, FRCSG, MS, MD, FRCP(London)

Deputy Editor

Ahmed Fahal

Deputy Editor

Reviewer's Responses to Questions

**Key Review Criteria Required for Acceptance?**

**Methods**

-Are the objectives of the study clearly articulated with a clear testable hypothesis stated?

-Is the study design appropriate to address the stated objectives?

-Is the population clearly described and appropriate for the hypothesis being tested?

-Is the sample size sufficient to ensure adequate power to address the hypothesis being tested?

-Were correct statistical analysis used to support conclusions?

-Are there concerns about ethical or regulatory requirements being met?

Reviewer #1: Are the objectives of the study clearly articulated with a clear testable hypothesis stated? Yes

-Is the study design appropriate to address the stated objectives?Yes

-Is the population clearly described and appropriate for the hypothesis being tested?Yes

-Is the sample size sufficient to ensure adequate power to address the hypothesis being tested?No

o

-Were correct statistical analysis used to support conclusions?Yes

-Are there concerns about ethical or regulatory requirements being met?No

**Results**

-Does the analysis presented match the analysis plan?

-Are the results clearly and completely presented?

-Are the figures (Tables, Images) of sufficient quality for clarity?

Reviewer #1: Does the analysis presented match the analysis plan? Yes

-Are the results clearly and completely presented?Yes

-Are the figures (Tables, Images) of sufficient quality for clarity? Yes

**Conclusions**

-Are the conclusions supported by the data presented?

-Are the limitations of analysis clearly described?

-Do the authors discuss how these data can be helpful to advance our understanding of the topic under study?

-Is public health relevance addressed?

Reviewer #1: -Are the conclusions supported by the data presented?Yes

-Are the limitations of analysis clearly described?Yes

-Do the authors discuss how these data can be helpful to advance our understanding of the topic under study?Yes

-Is public health relevance addressed? Yes

**Editorial and Data Presentation Modifications?**

Reviewer #1: None

**Summary and General Comments**

Reviewer #1: The authors present a cases serie report of paracoccidioidomycosis in HIV -infected patients. They describe the most relevant epidemiological , clinical and outcome features of these patients and compare them with the literature data . Due to paracoccidioidomycosis is another neglected disease in Latin American countries, the present report is very important in order to elucidate what is the real magnitude of this mycosis in the context of HIV infected individuals. Currently there is a great scientific interest to define if PCM can be added to the list of opportunistic diseases which define AIDS and its clinical presentation would be considered as another clinical form named mixt by some authors.

PLOS authors have the option to publish the peer review history of their article (what does this mean?). If published, this will include your full peer review and any attached files.

Reviewer #1: No

---

## [Editor Report · Acceptance letter]

10 Jun 2022

Dear Dr Marinho Falcão,

We are delighted to inform you that your manuscript, "Paracoccidioidomycosis in people living with HIV/AIDS: a historical retrospective cohort study in a national reference center for infectious diseases, Rio de Janeiro, Brazil," has been formally accepted for publication in PLOS Neglected Tropical Diseases.

Best regards,

Shaden Kamhawi

co-Editor-in-Chief

Paul Brindley

co-Editor-in-Chief
